# Power in the Context of SCM and Supply Chain Digitalization: An Overview from a Literature Review

Janosch Brinker * and Hans-Dietrich Haasis

Chair of Maritime Business and Logistics, University of Bremen, 28359 Bremen, Germany; haasis@uni-bremen.de
* Correspondence: jbrinker@uni-bremen.de

**Abstract:** *Background*: Within highly complex supply chain networks, driven by the trend of digitalization, supply chain relationship management becomes one of the central enablers in increasing supply chain performance. While the influences of globalization and digitalization on the supply chains are increasing, the power allocation within several markets is centralized to a small number of companies. The objective of this paper is to investigate the research gap concerning the impact of power asymmetries on the supply chain, in addition to the trend of digitalization. *Methods*: A literature review on power, in the research area of supply chain management and logistics, is used to synthesize the current state of the art in this research field and to provide a comprehensive definition of the concept of power. *Conclusions*: While this paper provides an overview of the impact of power allocations, according to supply chain digitalization and in the present research of supply chain management, it also develops a definition of Power in Supply Chain Management in general. Linked to this definition, this research elaborates on the research gap between power allocations and the digitalization of the supply chain.

**Keywords:** power; bargaining power; digitalization; innovation; supplier management; supply chain digitalization; supply chain management

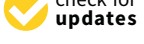



## 1. Introduction

Digitalization can be determined as one of the driving forces influencing inter-organizational relationships in the supply chain [1]. "The ability to combine massive data collection, previously unimagined information connectivity and visibility, and ever-improving analysis capabilities, combined with a physical network consisting of broad geographic network coverage, local fulfilment presence and parcel/postal delivery, have positioned these twenty-first-century retailers as leaders of the digital supply chain era" [2]. With the technological development, however, the cooperation of the actors within the supply chain [3] and the structure of the markets is changing [2,4]. Due to the multitude of players, the possibility of collaborative, cross-company cooperation is becoming increasingly important for the success of the supply chain in competition. Furthermore, the interaction of suppliers, customers and logistic providers on different tiers is creating a network of different patterns of dependency or influence. Depending on the network structure and other factors, such as company size, switching costs or resource dependencies, the ability of influencing is not equally partitioned along the chain, which leads to an asymmetry in the allocation of power according to every strategic decision which has to be taken along the chain [5].

Based on the growth of digital leaders and the increasing influence of digital technologies, market power within digitalized chains is increasingly centralized among a few players [4]. This change leads to a change in the distribution of power among the supply chain actors, in which small- and medium-sized enterprises, in particular, can become dependent on larger companies, so their own corporate goals have to take a secondary priority to those of their customer or supplier [6]. The increasing market share of companies such

as Amazon, ALPHABET INC or Alibaba can be used as a first indicator for this trend. In 2021, eight of the ten largest companies in the world were using digital business models [7]. Amazon also can be used as an example to describe the power shift, induced by its digital business models, which are restructuring the traditional structures in the retail sector and changing the power structures in the market by creating new patterns of dependency from the analysis of data or the implementation of data interfaces [8,9], as well as focusing the logistic infrastructure on a small number of warehouses and logistic infrastructures, instead of previous end-customer-orientated logistic processes. On the one hand, the analysis of customer data or data interfaces are creating new patterns of dependency and inducing a new level of resource dependency. On the other hand, the increasing market share of Amazon, based on their digital and centralized business model, allows Amazon to influence the decisions of their suppliers. In summary, Amazon is reaching a position of power, where it will be able to dominate its suppliers decision variables [10].

Digitalization projects, such as the implementation of block chain platforms from companies such as MAERSK and IBM or Walmart, which are creating a dependency on the data interface, data analysis and the fulfillment of the requested guidelines, are also representative for the structural changes in the value chain. The definition of guidelines and data interfaces furthermore creates market standards, which increase the level of dependency and market entry barriers. Further mechanisms can be illustrated in the industry standards of Microsoft OS. The increasing market share of Microsoft enables the company, next to the resealing of different software products, to define industrial standards and influence the whole IT infrastructure of several industries. So, server infrastructure, as well as its security and further IT processes and software applications, are often related to Microsoft OS [11]. The highly variable environment of digitalization is influencing the classic value creation processes and relationship structures [12]. In this increasingly complex environment, digitalization is inducing dependencies between different suppliers, whose effects are unidentified and thus represent an unidentified variable with regard to strategic decisions. Based on this development and the huge impact of buyer–supplier relationships on supply chain decisions [13], this research offers the first theoretical contributions for dealing with these challenges and complex transformation processes. To reach the research objectives of this review, the following research question was formulated: How is the concept of asymmetric power allocations discussed in the scientific literature, in the context of supply chain digitalization and supply chain management? According to the overall objective, this research closes the gap between the present supply chain digitalization research—which highly focused on technical implementation—and the research about supply chain relationship optimization.

## 2. Literature Review: Methodology and Descriptive Analysis

Referring to the previous section, the following sections will present the research method of a literature review about power, digitalization and innovation processes within the supply chain, in a comprehensible systematic and scientific structure, and it will present the current state of the art [14,15].

To generate a clearly structured review, the analysis is split into two main parts: (a) First, this research is developing a general overview of power in the research area of supply chain management and elaborating a definition of power in the research area of SCM. (b) The further research focus of the review is closely aligned to the title of the research and closes the research gap between supply chain digitalization and power allocations. The objectives of this review are to summarize state-of-the-art research about power in SCM, as well as the influences of power in supply chain digitalization, and to point out further research recommendations.

### 2.1. Methodology

The study is conducted according to Fink [14], who has proposed a seven-step, step-by-step approach, to generate a comprehensible scientific review. The different steps of the

analysis procedure are highlighted in Figure 1. The methodical approach is identical in both parts of the review, so the structure will only be described one time. Further considerations of this research will be carried out in addition to this split. The first step is the formulation of the research question, which guides the screening of the literature and is a superordinate of the review in the achievement of the goal [14].

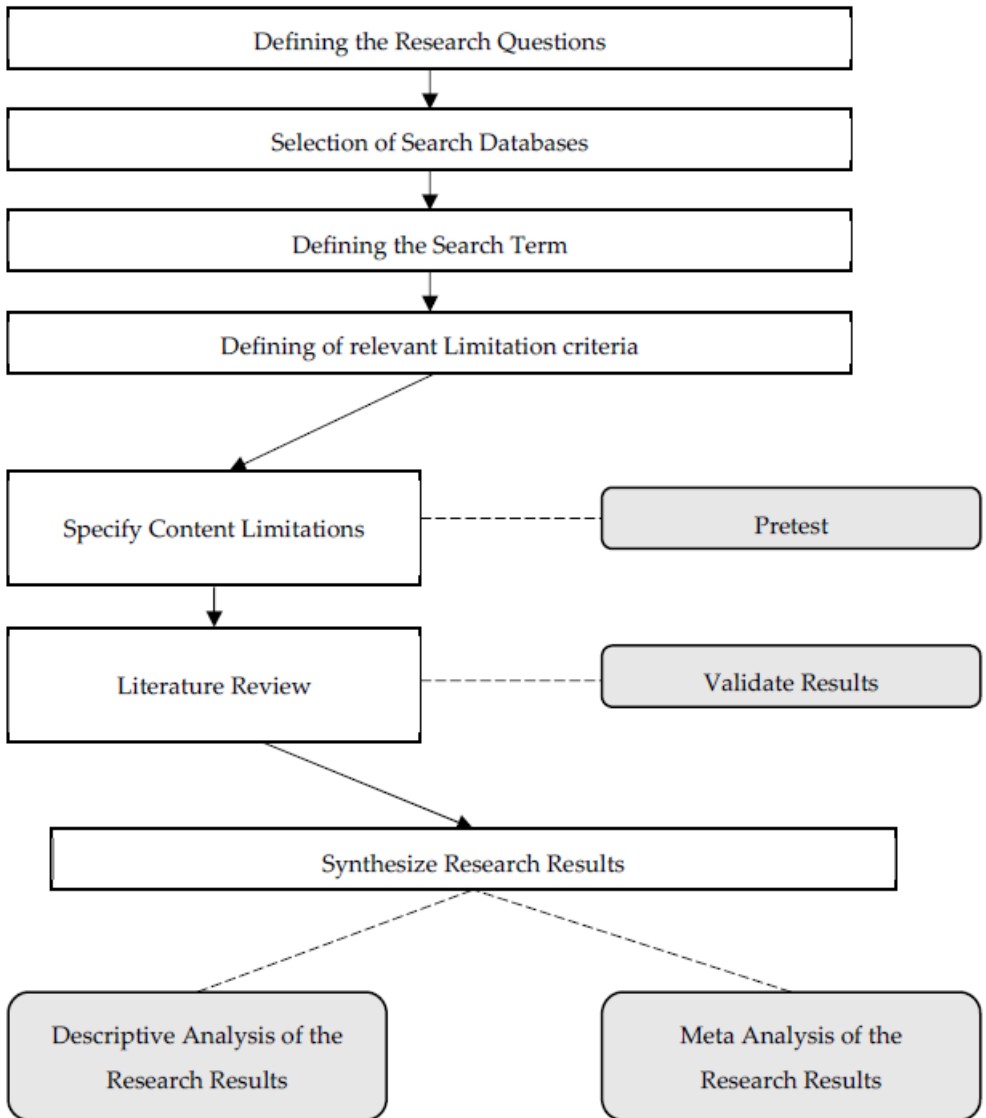

**Figure 1.** Steps of the literature review [14].

In addition to the objective of the research, three leading questions can be formulated:

1.  How is the term of power defined in state-of-the-art research on power within the supply chain?
2.  How does the existing research capture the impact of power on supply chain management?
3.  How does the existing research capture the impact of power on the digitalization of the supply chain?

Further relevant databases were defined and the publishing period was selected. The review was conducted in the scientific databases Web of Science and Scopus, which offers a wide scope of publications. Denyer and Tranfield [15] suggest that the search term is defined, which, in addition to Fink [14], is necessary for the localization of relevant literature, based on the theoretical background of the research area. The search terms, including synonyms of power, supply chain and digitalization, and all variants, are summarized in

Table 1. The different synonyms of each column are summarized by the logical operator of "and", and the synonyms of each row are connected by the operator of "or". According to the two different parts of this research, two different search terms resulted from this structure. In a first stage, only the first two columns of Table 1 were used, in a second stage, this was extended by the third column. As an example, for the database Web of Science, this results in the following search string: (TS = ("Supply Chain" OR "Supply Chain Management" OR SCM OR "Value Chain" OR "Supply Chain Relationship" OR "logistics") AND TS = ("Power" OR "Bargaining Power" OR "Buyer Power" OR "Customer Power" OR Legitimacy OR Reward OR Coercion OR Referent OR Informational)). This definition of the search term is fundamentally based on more general terms/synonyms of power, digitalization or supply chain to develop a wide overview about the research field. In doing so, different synonyms of power were defined on the basis of social researchers, such as Weber, French/Raven [16], Olsen [17], Popitz [18] or Sofsky and Paris [19], who developed a bright understanding of the term of power, whereby the main challenge for this review was the transformation of this understanding to the SCM area. Furthermore, the different synonyms of digitalization, as well as the synonyms of supply chain, were selected on a superficial level to not restrict the research by a technological focus and develop the broadest possible understanding. For example, terms such as Industry 4.0 or big data were deliberately excluded to reduce any technological focus from this part of the search term and offer a more general overview. In addition to the search terms, this review was limited to journal publications and conference proceedings which were published between 1990 and 2021 in the English language. The period of the last thirty years is, on the one hand, quite broad, but on the other hand, it offers a large framework, so both publications from the beginnings of supply chain management research and more recent research results about digitalization can be considered. The criteria of further limitations, except of the categorizations, are summarized in Table 2. In addition, both databases only consider publications that were published within the framework of an economic categorization, thus having a direct connection to supply chain management research.

**Table 1.** Definition of the search terms.

| Power | Supply Chain | Digitalization |
|---|---|---|
| bargaining power | Supply Chain Management | Digitalization |
| customer power | SCM | Innovation |
| buyer power | Supply Chain Relationship | Technology |
| legitimacy | Value Chain | Information |
| reward | Logistics | Automation |
| coercion | | |
| referent | | |
| informational | | |

**Table 2.** Criteria of inclusion and exclusion.

| Criteria | Inclusion | Exclusion |
|---|---|---|
| Document type | Journals, conference proceedings | Any other publications, such as reports or reviews |
| Publications stage | Final | Article in process |
| Language | English | Any other language |

Entering the search term into the selected databases, the research results highlighted 8574 hits for the database Web of Science in July 2021, and 22,661 hits were found in the Scopus database. These first results were narrowed down by using different levels

of limitations. For the Web of Science database, the search was narrowed down to the following categorizations: Management, Operations Research Management Science, Business, Economics, Business Finance, Transportation and Transportation Science. All other categories were excluded. Based on these restrictions, the search result reduced to 1481 hits. The same restrictions were made for the Scopus database. Here, the search result was limited to the categories Business, Management and Accounting, as well as Economics, Econometrics and Finance, and all other categories were excluded, which led to a search result of 953 hits. Other subject areas were excluded. A further narrowing based on an analysis of titles and abstracts finally reduced the search result to 68 publications in Web of Science and 46 results in the Scopus database. This step was based on a scanning of the title and abstracts according the any connection to the search term, based on qualitative reading. Finally, this result was considered with regard to duplicates and access authorization to the publications, so that 67 publications were considered for the final analysis.

A similar procedure was used for the specific literature analysis on power in the context of digitalization within supply chain research. In July 2021, the Web of Science database yielded 2004 hits, whereas Scopus yielded 5872 hits. The further restrictions regarding the categorizations are identical to the first for this second analysis. Thus, narrowing down based on categorization reduced the results to 361 results in Web of Science and 240 hits in the Scopus subject database, respectively. Further narrowing down of the results based on an analysis of titles and abstracts yielded a final result of three publications in Web of Science and three results in the Scopus database. These were also checked for duplicates and access permission, leaving four publications for the final analysis. If this review was carried out without the synonyms introduced for the term digitization (technology, information, automation, innovation), then two hits were found in the Scopus database and only three hits for the subject database Web of Science. Similar to the generally very low search result, this result also shows the so-far low extent of power research in the context of digitization. For further consideration, therefore, the synonyms of the term digitization were consulted, and the synthesized results were interpreted in the context of digitization.

The gradual limitation of the search result is summarized in Table A1. The further steps of the review (content analyses) are presented in the following sections.

*2.2. Descriptive Analysis: Characterising the Research Results and the Literature Surrounding Power in SCM*

After the selection of the relevant literature, the next steps comprised the detailed analysis of the literature [14,15]. In addition to the small number of publications in the research area of power and digitalization, a separate descriptive presentation was dispensed, and the results are presented summarized. As illustrated in Figure 2, the consideration of power structures has become increasingly important in recent years. The maximum number of publications was reached in 2017, with a total number of 13 publications per year. The overall number of publications between 2014 and 2020 was an average of 3.4 publications per year. Next to the small average number of publications, the increase in the number of publications that has be taken in recent years proves the previously low but increasing significance of the research area.

Figure 3 illustrates the number of publications in addition the publisher. The research result shows a huge number of publishers which have only published a small total number of publications. Only a few journals have published more than two publications per year. The maximum number of eight publications per year was accounted by the journal *Supply Chain Management*, and other journals, such as Industrial Marketing Management, European Journal of Operational Research and the Journal of Supply Chain Management, have published six papers. By taking into account this small number of publications per year, the huge number of publishers proves the low significance of power research in the research area of the supply chain.

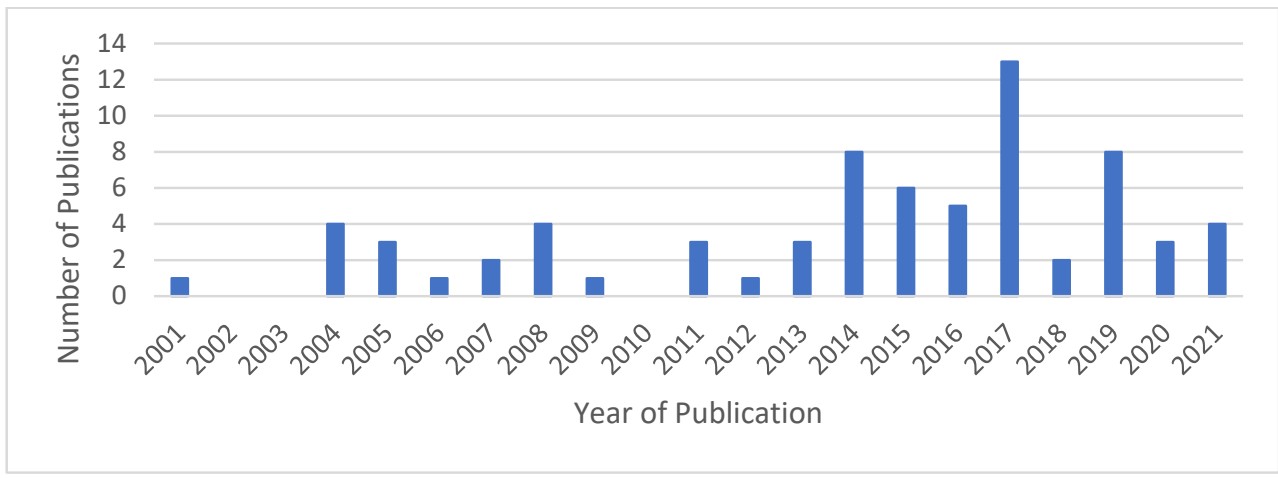

**Figure 2.** Number of publications per year.

**Figure 3.** Number of publications per source.

The following section will illustrate the content analysis in more detail. It will present a general definition of power in the research area of SCM and also giving an overview about the state of the art of the present research about the influence of power the digitalization of the supply chain. This analysis will have two parts: the presentation of the general review result, and the illustration of the overview about the impact of power on supply chain digitalization.

## 3. Power within the Supply Chain: An Overview

The following section presents an overview of the research about power in supply chain management and digitalization in SCM. To do so, this section is structured according to the three research questions, presented in the beginning of this paper. It starts with a general overview of the definitions of power used in the research subject and a development of a general definition of the approach to power in SCM, followed by a categorization of the results, ending with and an overview of the subjects of power and digitalization in SCM.

### 3.1. Development of a Power Definition Approach in SCM

According to this research objective, it was necessary understand the concept of supply chain power and to develop a general power definition in supply chain management. In a first stage, it can be summarized that authors are referring to the definitions of El-Ansary/Stern [20] and Emerson [21], or referencing the theoretical contributions of French and Raven [16]. To develop a more general definition of the approach to power, any publication of both reviews was analyzed according to the used definition of the term of power. Summarizing these considerations and refer it to SCM power in context of this research is defined as: *Power is the ability to influence the **decision variables** of other supply chain participants, based on a **mutual dependency relationship**, regarding to the **participants' individual preferences**.*

This definition forms the basis for all further considerations in this research. Power is based on an individual ability, which is mostly based on a resource dependency, to influence decision variables of other supply chain actors, towards which power is applied in order to promote the achievement of one's individual cooperate goals [22–34]. The goals of this exertion of influence can range "from quality and delivery requirements to prices and contractual terms, through to issues of strategic direction, product development and competitive intelligence" [28]. The ability to exert influence here is anchored in the relationship between the actors and is rooted in the use of different influence strategies. An overview of the different influence strategies and influence mechanisms is outlined in Table A2 According to this definition of power, equal and symmetric relationships cannot be characterized by power. The term of power itself requests an asymmetric allocation in the ability of influence [5]. Referring to this, an asymmetric distribution of power can be understood as the unilateral capability of influence, so one actor in a dyad is in position to reach its individual goals by influencing the decision variables of the other.

According to the power bases theory of French and Raven [16], these influence strategies can be subclassified by mediated and non-mediated power strategies. Mediated power strategies are here considered to be constraints or legally based legitimations. Non-mediated power strategies include information power, expert power and referential power [22,35–38]. Following Benton and Maloni [22], the influence mechanism of reward should be listed separately under the categorization of reward-mediated power.

The analysis of the publications proved that a large part of the influence strategies is based on a resource dependency of the respective partners. Thus, 24 of the publications are directly related to the resource dependency theory (RDT), according to Pfeffer and Salancik [39]; in addition, other publications also describe an influence strategy that is based on resource dependency, which is described as a mechanism of influence with regard to the influence strategies coercion, reward, informational power and expert power. Legitimate power and referential power, on the other hand, are not justified on the basis of this dependence.

In addition to resource dependence, various authors refer to the incurrence of transaction costs or transaction cost economics (TCE) as an influential mechanism with regard to various influence strategies. Ireland [40] cites the creation of switching costs and the resulting costs for the change of supplier. Cox et al. [41] describe the occurrence of transaction costs as part of a negotiation process and link the goal of economic action with the avoidance of such costs. Based on these costs, they also refer to the number of alternatives and the development of switching barriers [22,24]. Furthermore, TCE can be found in the evaluation of the supplier relationship [23]. In addition to the RDT and the TCE, few other mechanisms of influence can be found in the literature. Ref. [42] referring to the influence of power through process integration and the resulting ability to influence decision-making processes through information/expert knowledge. Wang et al. [43] cite the same definition of power, but break down influence into market dominance and channel dominance. The resulting influence strategies increasingly refer to influencing price or channel strategies and are, however, also linked to the use of financial resources.

In summary, it can be highlighted that the term of power is composed on three different levels: the definition approach itself, influence strategies and influence mechanisms. The Figure 4 highlights these different dimensions of power, which could be used as a theoretical approach to analyzing the influences; for example, digitalization can have on the power allocation of the supply chain. Furthermore, this first analysis can highlight the variety of different definition approaches and the necessity to develop a more SCM-specific and empirically validated definition. This research offers a first concept, which can be used for further validation.

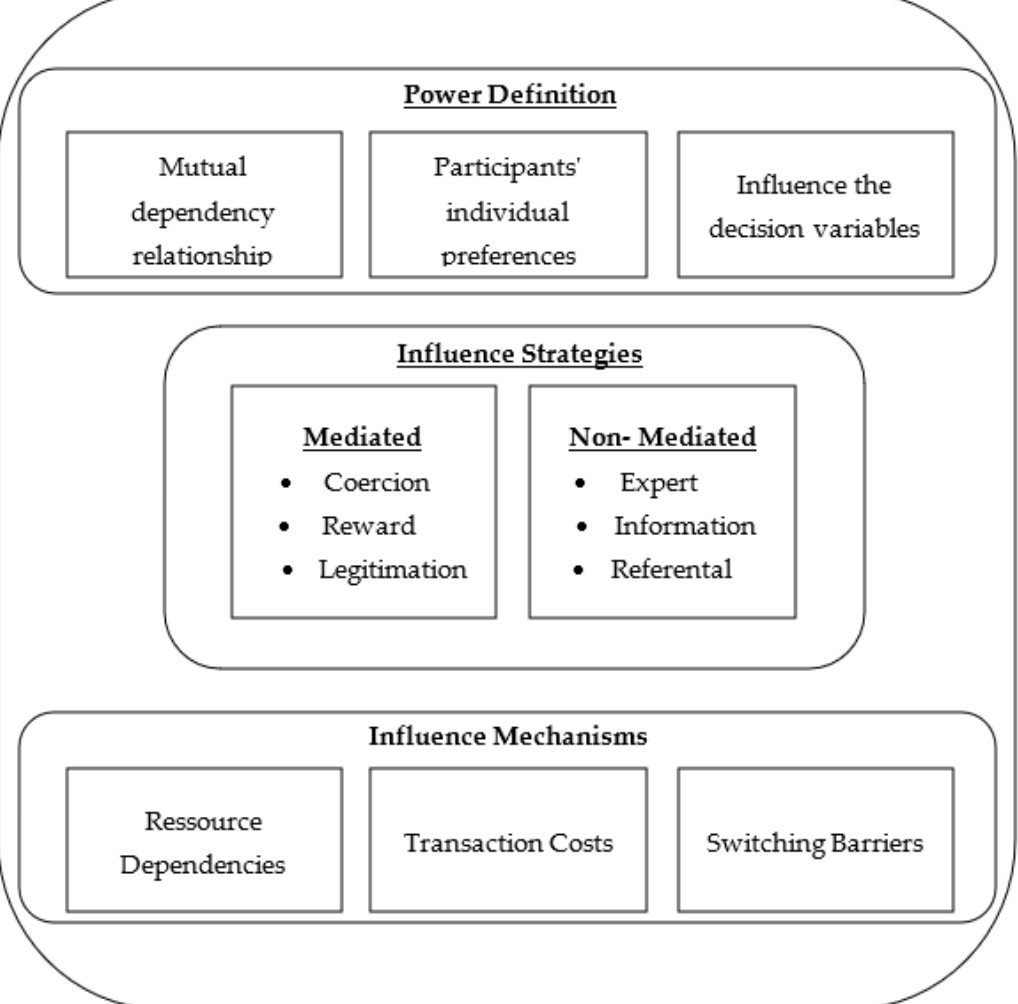

**Figure 4.** Dimensions of power.

*3.2. Power as a Research Subject in SCM*

According to the objective of this review, the results will be categorized and outlined in order to provide an overall picture of the research topic. Six superordinate categorizations of the research result, which are developed based on the research topics of each publication, can be highlighted. Every publication was assigned to the category it matches the most, which led to an overlap in some categories content. These focal points are presented in Table 3, which also presents the main keywords of this category.

Thus, various definitions of power repeatedly reference a resource dependency as the basis of individual power, although the respective emphases of the individual studies are set differently. The aspect of opportunism was a recurring element in the power research. It can be found in the considerations about bargaining power by Sheu and Gao [32] or Fabbri and Klapper [44], but was also taken up in the analysis of power strategies of the construction industry in the UK by [45], who linked power with the term of individual opportunism—the ability to assert one's individual goals in the face of resistance from others. An opportunistic behavior is, in this context, understood to be very closely connected, with a minimization of the total benefit, which can be seen as the cumulative benefit of all parties [46].

**Table 3.** Categorization of the research results.

| Categories | Keywords | Author |
|---|---|---|
| Bargaining Power in the Supply Chain | Bargaining power; Dominance; Channel coordination; Negotiation; Buyer–supplier negotiation; | [25,27,32,47–49,59,60] |
| Power Structures along the Chain | Power Structure; Power dominance; Power Asymmetry; Chain Structure; Business networks; Influence Strategies; Resource Dependence Theory; Control power; Principal–Agent Paradigm | [29,42,45,50–54,61–69] |
| Influences of Supply Chain Design on Power Structures | Channel Dominance; Competition; Global sourcing strategy; Relationship initiation; Supplier attractiveness; Supplier selection; Channel selection | [55–57,70–73] |
| Strategies of Power and Supply Chain Relationship Management | Relationship; Relationship marketing; Supplier satisfaction; Power use; Sustainable supplier management; Customer value; Supplier value; Buyer–seller relationships; Channel relationships; Power; Power-imbalanced relationships; Market power; Power-dependence management | [22,24,28,30,33,34,36–38,46,58,61,74–81] |
| Collaboration and Power | Collaboration; Contractual Governance; Relational Governance; Supplier relationship; Dependence; Supplier orientation; Logistics triad | [78,82–87] |

As well as the negative effects of an opportunistic behavior, the superordinate focus of symmetric power allocations in the chain can be highlighted in several publications and categorizations. Concerning the construct of power, the influences of opportunism and the influence of symmetric power allocations, there are only minor differentiations between all publications; furthermore, these categorizations are overlapping in several considerations, theories and results. However, differences can be found in the respective approaches to the topics of power, within the supply chain.

In a first superordinate category, the literature highlights an analysis of bargaining power or negotiation power in the supply chain. Based on the derivation of the bargaining power, different manifestations on the supply chain and the enforcement of individual goals can be justified. This category overlaps in content with the categories of power structures and power strategies discussed below. The availability of resources is associated with the bargaining power of individual partners in the supply chain and will have an influence on the negotiation process. Hereby, resources include material, financial resources and even the availability of information [25,27]. Another possible bargaining position can

be generated by the creation of switching barriers. The respective negotiation strategy, the choice between reward or coercion strategies, has a decisive influence on the success of a negotiation situation. The enforcement of corporate goals, such as the enforcement of a pricing strategy, is positively influenced by the respective bargaining power of the contracting parties [47,48]. The transparent sharing of one's own negotiating position reduces the negative effects that can result from premature action [49].

The analysis of different dominance structures within dyads along the chain pointed out that opportunistic behavior will negatively influence the supply chain. Mutual dependencies between respective partners or a shift of power structures towards the customer side reduces opportunistic influence, which leads to a minimization of the total benefit [50]. Similar to the collaborative approach, the highest benefit is reached by a symmetrical distribution of power. Symmetric power relationships and strong ties between the parties can foster the will of individual suppliers/customers to invest in expanding these relationships and increasing total benefits [29,51]. Individual dominance strategies increase opportunistic benefits, but simultaneously it can negatively influence the supply chain's overall success [51]. In addition to the positive influence of innovative solutions—in this context understood as IT solutions—Deitz et al. [52] used this example by analyzing the influence of forced IT integrations. The empirical analysis shows the positive correlation of company liquidity on the implementation of forced measures, such as the introduction and integration of RFID chips in production. This power allocation can only be narrowed down for isoelastic demand, which induced a shift of the distribution of power towards the traders. In non-price-sensitive markets, in which the customer is strongly dependent on product availability, the total benefit can be increased more strongly by, e.g., pricing on the part of retailers than by process optimization on the part of manufacturers [53,54].

The analysis of different market structures or supply chain designs and their influence on power structures and supply chain performance forms the third category, which shows that the individual benefit can be maximized through an appropriate power position [55]. For example, a power position of the manufacturer can lead to an increase in production volume, a decrease in the "retail price and the largest expected surplus for an individual buyer" [56], on the other hand, the total "channel profit and the total consumer surplus" [56] can increase due to a power position of the retailer [56]. So, the market power structure will have a noticeable influence on the resale pricing strategy within multichannel service supply chains [57]. Due to the influence of the more powerful actor, the objectives and target achievements of the entire chain continue to differ, but the chains targets are often influenced by the most powerful member.

In addition to examining the power structures within the supply chain, further publications focus on the analysis of power strategies or strategies of exerting influence. These considerations also take up the investigation of the relationships along the supply chain and their influence on the performance of the supply chain. Although there is a major overlap with the previous section in terms of content, these publications differ in their focus on relationship management and power strategies. Cox et al. [41] highlight the meaning of individual power strategies, leaned against the power base theory, and a corresponding resource dependence. They emphasize the complexity of the various supply chain structures and suggest that each power strategy must be evaluated to the background of the individual requirements. Individual power strategies exhibit varying degrees of positive or negative effect on the supply chain relationships. Strategies of coercion, overriding mediating strategies, can significant negatively influence supply chain relationships, whereas non-mediating strategies have a significant positive effect. Strategies of reward turn out to be largely positive in terms of relationship quality within the supply chain and thus performance, but do not show a significant impact [22,25,37,58]. Assuming that power structures exist within each dyad, Maglaras et al. [34] and Takashima and Kim [24] show that corresponding factors of asymmetry negatively influence the respective relationship structures and performances, as well as the negative effect incompatibility of one's own objectives with those of the respective partners will have on the success of the supply

chain by the mechanism of dependency, whereby Chicksand and Rehme [46] refer to value creation, which is based on the individual objectives and power strategies. Contrary to this, Hingley [58] point out that power is not to be seen negatively throughout, but asymmetries are present as a basic condition of all interactions, and only the use of a dependence leads to negative effects.

As part of supply chain relationship management, the factor of power is an effective tool for managing suppliers and customers. Symmetrical power relationships can be highlighted as one driving forces to maximizing benefits along the supply chain. Considering collaboration and the effects of symmetrical power structures within the supply chain, forms another, overarching research focus. Several publications take up the collaborative approach to supply chain design, in the context of analyzing power effects on supply chain performance. Referencing the power mechanisms Benton and Maloni [22], the literature pointed out the positive influence mechanisms of non-mediated power strategies on supply chain success. Similar to non-mediating power strategies, which positively influence supplier relationships quality, a collaborative strategy also has a positive effect on supply chain performance by aligning strategic goals [86]. Collaboration, however, does not change the power structures in the chain, rather it creates a shared decision horizon and joint strategies, in which decisions are made for mutual benefit. Resource dependencies, the monetary strength of individuals and the asymmetric information relationships remain, but can be softened with regard to the joint achievement of goals. Therefore, it is necessary to know about one's own individual market position, the market as a whole and possible influence strategies, and to include this knowledge in the strategy development [36,61,62].

The final categorization forms the element of trust and its influence on power structures and supply chain performance. This category increasingly overlaps with the topic of collaboration. The recognition of an authority increases the confidence in the respective power position, from which an increase in the respective influence results, whereby this is not regarded as negative or opportunistic. Creating a common identity has a similar effect. This also increases trust among the different supply chain members and strengthens the power structures along the supply chain. Inter-organizational interfaces also have a trust-building effect; for example, they promote the exchange of information and thus increase the degree of trust and the associated benefits for the supply chain. Finally, perceived equity is asserted as an influencing factor on trust within the supply chain by [88]. According to supply chain governance approach, the level of trust is positively influenced by the degree of information sharing and thereby increases the common benefit. Trust here represents the basis for creating long-term strategic partnerships, across the presence of asymmetrical power distributions [26,89]. The more trustworthy a supplier/customer is, the better the individual's respective competitive position. The size and market power of a company increases the corresponding trustworthiness [35].

### 3.3. Power and Supply Chain Digitalization

The following section analyzes the reviewed literature, linked to the research area of SCM and digitalization. The concept of supply chain digitalization in this review is associated with technological improvements, such as IoT, CPS and smart products, as well as being driven by technologies such as big data, etc.; however, to avoid technological specialization, this review focusses on general terms, such as digitalization, etc. Including the preceding considerations about supply chain digitalization, the collection and analysis of data will become a critical resource in the chain. Overall, it can be concluded that the concept of power is only addressed in a very rudimentary way in relation to supply chain digitalization. Several publications are dealing with the impact of information availability on collaboration or relationship quality. Further results can be mentioned in the impact of new business models, such as e-Commerce, on supply chain structures and supply chain design. Based on the induced restructuring of the chain and value creation, first approaches about the relation between digitalization and supply chain power structures can be elaborated.

Based on this assumption, information sharing can be used as an influence mechanism of power; furthermore, it will have an impact on the level of collaboration, by influencing the level of trust. Vendrell-Herrero et al. [10] hereby lead a bidirectional perspective on the theorem of power: digital servitization can strengthen the power position of downstream companies, if they gain control over the connection channels to consumers; furthermore, a strengthening of the position of upstream companies can be demanded by regaining a resource dependency. In addition to the use of information as an influence mechanism of power, Vendrell-Herrero et al. [10] link the concept of digital disruptive technologies and business models to the analysis of enterprise collaborations within supply chains. The use of digital technologies enables companies to introduce new products or distribution channels, leads to a change in value creation, such as corporate processes, and ultimately changes competition in the market [8,90]. By considering the availability of information and the capability of data analysis, data processes have become another central resource of the chain, which influences the power allocations.

Referring to the principal–agent theory, power asymmetries in the chain can be deduced in dependency to information's availability. The allocation of information is asymmetric allocated on side of the agent, who can decide which information it is willing to share with the principal. However, informational availability or expert knowledge can be understood as mechanisms of coercion or reward. Information sharing, depending on the individually chosen corporate strategy, can on the one hand promote the consolidation of collaborative strategies, but on the other hand, only the withdrawal of information availabilities or knowledge can be used as a strategy to expand asymmetric power distributions [89]. As an example, this development in value creation, induced by digitalization, can be located in the more original, end-customer-oriented sales landscape of stationary retail, where the final customers make their procurement decisions in the stationary store, which also takes over the functions of the warehouse logistics. One of the main objectives of the stationary trade concerns the reduction in transaction costs. This creates a corresponding power position of stationary retailing—on the one hand, there is a customer proximity, and their purchase behavior can be analyzed, on the other hand, further instances and tasks of the supply chain takes over. With the growth of e-commerce platforms or mobile shopping, the supply chain expands to include another actor. The power structures within the chain shift accordingly. Like brick-and-mortar retail, e-commerce platforms meet the needs of the end customer and expand their market share. Similar to stationary retail, the platforms receive information about the customer's preferences, but at the same time they reduce transaction costs on the customer side through significantly increased transparency and comparability. Distinctions are primarily found in logistics—the transport of products to the customer. In addition to the reduction in transaction costs, they point out various influencing factors that distinguish digital commerce from analog commerce and establish added value for the customer. Furthermore, they cite customizability, spatial independence and corresponding interaction as added value. In these factors, Reinartz et al. [9] justify the increase in power that digital sales models gain over analog sales models.

Digitalization leads to changes in value creation and competition, further to a growth of digital monopolies, and so it is influencing the allocation of power in the chain [90]. Similar to the approaches of Reinartz et al. [9], Subramaniam [90] shows that digitalization creates value when processes or collaborations can be made faster and more easily due to digital measures, thus creating a competitive advantage in the market. Market entry barriers, such as physical availability of products and raw materials or product developments, are no longer the only limiting factors. The analysis and availability of data is becoming increasingly important. Following Subramaniam [90], it can result in a monopoly position that is both product-associated and data-driven. With regard to the consideration of digital monopolies, he concludes that the original definition of a monopoly, tied to the respective market shares, is no longer sufficient.

## 4. Concluding Discussion

The objective of this research was to analyze the scientific literature according to the merge of power in SCM and the research field of supply chain digitalization; therefore, assertions about the influences of asymmetric allocations of power, in addition to the trend of digitalization of the supply chain, can be made. The accomplishment of this objective is based on three basic elements: by developing a general definition of power, this research identifies RDT and TCT as the main influencing strategies of power on supply chain power allocations. Digitalization changes the competitive environment and the channel structures, so it can lead to emphasized asymmetries of power. Thereby, information technologies offer opportunities to reduce barriers in communication or information sharing.

### 4.1. Theoretical Implications

This literature review pointed out that research concerning objective power is becoming increasingly important in supply chain research, but these results are often theoretical and based on model assumptions. The research field of power overlaps with the research of collaboration, supply chain relationship management and supplemental, often simulative, game theoretical analysis about the impact of different dominance structures in supply chain management. The present research mostly focused on the influence of opportunistic behavior, not the influence of power structures, and its impact to the total supply chain benefits. Further superordinate research focusses can highlight the effects of asymmetric power allocations and how these will influence supply chain design or strategy. Thus, the research indicates a clear objective regarding the monetary influences that power can have on the supply chain. This review provides the first overview highlighting a few research gaps.

Basically, on this first level, the necessity for developing a general, SCM-specific definition approach about power and proving it on an empirical level can be highlighted. The present research offers a huge variety in the number of definitions and matches with other research areas in SCM.

In summary, the influences of asymmetric power allocations and strategic considerations are rarely considered within current research, but can derived from the influences of opportunism or symmetric power allocations. On this second level, the research about power in SCM is developing, but most results are grounded in the considerations of collaboration or opportunistic behavior, so there is also a lack of empirical confirmation and further theoretical research. Further research should to prove the results and definition approaches of this review on an empirical level, or could use the results presented here for some theoretical improvements. The main problem for this research area will be the delimitation of the concept of power in order to be able to elaborate actual further results. Possibilities for this are likely to be provided by qualitative research.

Thus, in this second level, it can be highlighted that the concept of power is frequently used, but is not detailed elaborated and empirically proved; accordingly, however, the first theoretical results overlap with collaboration, etc., so this can be used as a foundation for further concepts.

Although informational power is already used as one of the power bases of French and Raven [16], the influences of digital technology and its disruptive influence on the power structures of the supply chain are only superficially elaborated. The trend of digitalization and the changed competitive environment is considered but is only researched on a rudimentary level.

Further considerations of the influence of digital power and the derivation of strategic implications are completely lacking within these discussions. In particular, further research can, for example, focus on the influence of digitalization on the different levels of the power term, which has been developed in this paper. It will be the main challenge to deduce how the digital environment influences the different dimensions of power in SCM. Table 4 highlights the various research gaps that can be identified in this review, and further its outlines a basic research framework.

**Table 4.** Research gaps concerning power in SCM.

| Research Focus | Research Gap |
|---|---|
| Generalizable SCM Power Definition | Empirical validation |
| Influence Strategies | Empirical validation of Power bases in SCM |
| Influences of Power on SCM Strategy | Theoretical contributions and empirical validation |
| Similarities of Collaboration/Opportunism and Power research | Theoretical contributions |
| Power and SC digitalization | Theoretical contributions and empirical validation |

### 4.2. Managerial and Political Implications

From a management point of view, the knowledge about the influences of different power structures on the chain can be helpful tool for risk or resilience management and the strategic supply chain design. As shown before, the trend of digitalization is inducing huge changes in the markets and processes. In this more complex, competitive environment, every actor must deal with an increasing amount of information, interfaces and increasingly complex requirements of the market. The knowledge about these challenges and influences of power will improve the performance of every single actor and the whole supply chain. So, it can be useful to analyze the companies and the supply chain perspectives separately. On the company tier, it will be necessary to define your own market position and influence variables, which other companies can use to influence your decisions. This information will help you to increase your market share and decrease the influence of other participants on your behavior. Additionally, the knowledge about the individual long-term company goals will be helpful to deal with these market challenges. From a supply chain point of view, the knowledge about the power structures, different individual goals and the overall supply chain goal will be helpful to improve the overall supply chain benefits and create a long-term-orientated chain. Opportunistic influences of individual companies can be reduced or blocked, reducing the opportunistic behavior which can be implemented.

### 4.3. Limitations of the Research

The objectives of this research were to investigate the research gap concerning power in supply chain research in the context of digitalization and to develop a research framework for further research. To do so, a literature review was conducted. Fundamentally, the term of power in SCM is only used on a superficial level and defined qualitatively. In a first stage, it can be highlighted that the results about power in the state-of-the-art research are characterized by a degree of fuzziness, which negatively influenced these results; furthermore, it is one of the main objectives of this research to develop a general SCM-specific definition approach to decrease this degree of fuzziness for some further research. Furthermore, the definition of a search term for this review is influenced by this degree of fuzziness, as well as the lack of empiricism and the fact that all results are based on theoretical constructs reinforces this impression.

This review is primary limited by the subjective influences of the researchers in the selection of the search term and the limitation criteria, as well as the degree of fuzziness in the definition of the term of power in the overall research and the selection of relevant papers based on abstracts/titles. Fundamentally, the search term selection is based on the systematic classification of the subject area in the scientific foundation, but this classification is characterized in its design by subjective influences of the research process and the researchers. Based on this fact, as shown in the results and in the fact that most publications in this subject refer to just a few of the same theoretical contributions, this subjective influence can be minimized. All these limitations and categorizations have to be reviewed according to these possible subjective influences, and have to be verified in further research. Further research must verify the selection process of the papers to increase the objectivity by, for example, using larger groups of researchers and carrying out pretest validations.

**Author Contributions:** Conceptualization, J.B. and H.-D.H. methodology, J.B.; formal analysis, J.B.; writing—original draft preparation, J.B.; writing—review and editing, J.B.; supervision, H.-D.H. All authors have read and agreed to the published version of the manuscript.

**Funding:** This research received no external funding.

**Institutional Review Board Statement:** Not applicable.

**Informed Consent Statement:** Not applicable.

**Data Availability Statement:** Not applicable.

**Conflicts of Interest:** The authors declare no conflict of interest.

## Appendix A

**Table A1.** Research results per limitation stage.

| Review 1 | | |
|---|---|---|
| | **Web of Science** | **Scopus** |
| First research results | 8574 | 22,661 |
| Limitation based on categorizations | 1481 | 953 |
| Limitation based on title and abstract | 68 | 46 |
| Limitation based on accesses and duplicates | 67 | |
| **Review 2** | | |
| | **Web of Science** | **Scopus** |
| First research results | 2004 | 5872 |
| Limitation based on categorizations | 361 | 240 |
| Limitation based on title and abstract | 3 | 2 |
| Limitation based on accesses and duplicates | 4 | |

**Table A2.** Influence mechanisms and strategies of power.

| Author | Resource Dependency Theory | Further Influence Strategies |
|---|---|---|
| [76] | X | Transaction Costs Theory |
| [45] | X | |
| [22] | | Transaction Cost Theory/Number of Alternatives |
| [23] | | Transaction Costs Theory |
| [58] | X | Transaction Costs Theory |
| [25] | X | |
| [88] | X | |
| [26] | X | |
| [86] | X | Transaction Costs Theory |
| [65] | X | |
| [81] | | |
| [32] | X | |
| [91] | X | |
| [89] | X | |
| [33] | X | |
| [66] | X | |

Table A2. *Cont.*

| Author | Resource Dependency Theory | Further Influence Strategies |
|---|---|---|
| [35] | X | Transaction Costs Theory |
| [36] | X | |
| [24] | X | Transaction Cost Theory/Number of Alternatives |
| [47] | | |
| [74] | X | |
| [63] | X | |
| [82] | X | |
| [38] | X | |
| [64] | X | Transaction Costs Theory |
| [46] | X | |
| [83] | X | |
| [70] | X | |

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
