# Peer review of "Power in the Context of SCM and Supply Chain Digitalization: An Overview from a Literature Review"

_logistics, 2021_

Round 1
Reviewer 1 Report
- The title of the paper is not appropriate at all and is confusing.
- The keywords section should be revised. There are many keywords in this section. Please select the most appropriate ones.
- The reference style of the paper should be revised. Please use MDPI reference style.
- In the introduction, the authors suddenly started with “power”, I do believe this is very confusing for a reader that is going to read this paper. The authors should start with motivations for this study and the significance of this study.
- Line 108, please use “section” instead of the chapter.
- The research methodology has been defined appropriately, but some corrections are required. For example, the authors should incorporate table 3 in section 2.1, not the appendix. In addition, a table should be added to summarise inclusions and exclusions, see: https://doi.org/10.1016/j.ijdrr.2021.102627 , in addition, I suggest that the research methodology be summarised in a flowchart.
- Please fix the problem in line 149.
- The biggest problem of this paper is lacking using appropriate tables to summarise the contents of the literature review. It is very hard for a reader to find the research gaps in the current research area and potential directions for future studies.
- The current gaps have not been significantly highlighted.
- The limitation of this study should be discussed.
Author Response
- The title of the paper is not appropriate at all and is confusing.
- We have changed the title to a more appropriate title.
- The keywords section should be revised. There are many keywords in this section. Please select the most appropriate ones.
- Keyword section is updated
- The reference style of the paper should be revised. Please use MDPI reference style.
- Reference Style is updated
- In the introduction, the authors suddenly started with “power”, I do believe this is very confusing for a reader that is going to read this paper. The authors should start with motivations for this study and the significance of this study.
- We have improved the introduction to make it a little shorter and highlight the significance according to our review.
- Line 108, please use “section” instead of the chapter.
- Fixed
- The research methodology has been defined appropriately, but some corrections are required. For example, the authors should incorporate table 3 in section 2.1, not the appendix. In addition, a table should be added to summarise inclusions and exclusions, see: https://doi.org/10.1016/j.ijdrr.2021.102627 , in addition, I suggest that the research methodology be summarised in a flowchart.
- We have improved this section by elaborating the methodical approach and including a few tables and flowcharts.
- Please fix the problem in line 149.
- fixed
- The biggest problem of this paper is lacking using appropriate tables to summarise the contents of the literature review. It is very hard for a reader to find the research gaps in the current research area and potential directions for future studies.
- The current gaps have not been significantly highlighted.
- We extended the discussion of the results and include a few annotations according to the research gaps.
- The limitation of this study should be discussed.
- Discussion is included.
Reviewer 2 Report
File with comments is attached

Author Response
"The objective of this paper is to develop a research framework and investigate the research gap about the impact of power asymmetries on the supply chain, in addition to the trend of digitalization. This paper provides an overview of the impact of “power” according to supply chain digitalization and in research of supply chain management, also develops a definition of power in SCM in general.
The article focuses on Literature Review of a term linked to supply chain. The review topic could be considered as normal research topic to explore what previous scientists try to address.
The aim is to identify the gaps, related to the review topic, and according to our review, the goal has been accomplished.
From this reviewer's point of view, the effort to improve could be done to highlight the work done if needs it."
We improved the whole paper according to your remarks: included a few figures and tables to improve the methodical approach, extended this section by some discussions of the search term etc. Further, we highlighted the gap in the concluding section, as well as in the content analysis section (chapter 3&4). Finally we added a the research limitations to this review.
Reviewer 3 Report
Your introduction part is a bit too general. It can be shorter but more direct to the point.
The methodology section is confusing and needs major improvement. Details of the work should be clearly presented so the next scholar can repeat the work and get a similar outcome. It should be separated from the result presentation
I am not sure if the term 'systematic literature review' is suitable in this paper or not. Maybe a PRISMA approach makes it a better match. For example, what kind/language of publication was considered or if any filtering procedure was performed due to lack of article accessibility or etc.
Figures' details are not printed properly so some parts are missing
Structuring of the report needs adjustment with better writing and sectioning.
search terms needs discussion and crtical evaluation to make sure nothing is missing or over-highlighted.
I sense a little overstatement in some presentations that needs to be toned.
Author Response
Your introduction part is a bit too general. It can be shorter but more direct to the point.
- We improved the introduction and highlighted the significance of our research.
The methodology section is confusing and needs major improvement. Details of the work should be clearly presented so the next scholar can repeat the work and get a similar outcome. It should be separated from the result presentation.
- The methodical part of this review is improved by extending this section, including flow charts and including tables for the limitations. The PRISMA approach is kind of similar to the review approach of Fink (2014), but Fink offers more general guidelines for a review, which is more suitable to the kind of qualitative term of power, which is discussed differently in many research areas.
Figures' details are not printed properly so some parts are missing
- Improved
Structuring of the report needs adjustment with better writing and sectioning.
- We restructure the review in a few parts (methodical approach, content analysis etc.) and added a few sections to improve the readability of the review.
search terms needs discussion and crtical evaluation to make sure nothing is missing or over-highlighted.
- We include a small discussion and critical reflection of the search term, as well as adding your remarks to the limitations of this research, we include in the last section.
I sense a little overstatement in some presentations that needs to be toned.
- Improved the formulations and revised the review according to your sensed overstatements (for example introduction).
Round 2
Reviewer 1 Report
Accept
Author Response
Thanks for your revision and your remarks.
Reviewer 3 Report
The author has enhanced the article, but there are still some areas that require further work.
The reference styles are perplexing.
The presenting structure could be better.
The method section still needs to be improved in terms of providing detail, for example, how are the search keywords listed?
Still, to me, the term "systematic review" does not seem to fit this paper and should be altered or removed.
Why Industry 4.0 is not used for the digitalization category ?
Author Response
The author has enhanced the article, but there are still some areas that require further work.
- I placed a MS Word file in the attachment using the “Track Changes Tool" of MS office so it it easier for you to track all changes that has be made.
The reference styles are perplexing.
- We checked and improved the reference style.
The presenting structure could be better.
- We restructured the review according to the presented research questions, (summarized sections 3 and 4).
The method section still needs to be improved in terms of providing detail, for example, how are the search keywords listed?
- We added some more detailed explanations about the methodical approach to this section. One main problem of this kind of qualitative research is the subjective influence of the reviewer in the selection process, which we also add in the limitations.
Still, to me, the term "systematic review" does not seem to fit this paper and should be altered or removed.
- The term "systematic" is removed from this review. We double-checked Fink (2014) and Denyer/Tranfield (2009) to make sure not to miss any imported part of their methods: Denyer/Tranfield provided the term of an systematic review, but according to the fact, that our review is mainly based on Fink (2014) is is the right suggestion to exclude the "systematic".
Why Industry 4.0 is not used for the digitalization category?
- We decided to only include very general, superficial terms of digitalization to the search string and not focus too much on any technological impact. To make sure that we have not excluded any important papers we extended it would be possible to extend the search term for example by four main areas of the digital supply chain management (industry 4.0, Big Data, adaptive manufacturing and tracking or tracking technologies). The extension of the search string by for example industry 4.0 will extend the first results by 22 papers, but none of them is related to the term of power if we checked the abstracts.
Round 3
Reviewer 3 Report
Still, some minor adjustments are required. For instance, the title you chose is a little puzzling. I also notice some textual and formatting errors. You may, however, edit them later based on the advice of the publishing editor. Congrats!